# Label-Free Detection of Saxitoxin with Field-Effect Device-Based Biosensor

**DOI:** 10.3390/nano12091505

**Published:** 2022-04-28

**Authors:** Najeeb Ullah, Beenish Noureen, Yulan Tian, Liping Du, Wei Chen, Chunsheng Wu

**Affiliations:** 1Institute of Medical Engineering, Department of Biophysics, School of Basic Medical Science, Xi’an Jiaotong University Health Science Center, Xi’an 710061, China; ullahnajeeb@stu.xjtu.edu.cn (N.U.); beenishktk@stu.xjtu.edu.cn (B.N.); cnyulantian@stu.xjtu.edu.cn (Y.T.); duliping@xjtu.edu.cn (L.D.); 2Department of Biochemistry and Molecular Biology, School of Basic Medical Sciences, Xi’an Jiaotong University Health Science Center, Xi’an 710061, China

**Keywords:** electrolyte-insulator-semiconductor sensor, aptamer, PAMAM dendrimer, saxitoxin, marine toxins

## Abstract

Saxitoxin (STX) is a highly toxic and widely distributed paralytic shellfish toxin (PSP), posing a serious hazard to the environment and human health. Thus, it is highly required to develop new STX detection approaches that are convenient, desirable, and affordable. This study presented a label-free electrolyte-insulator-semiconductor (EIS) sensor covered with a layer-by-layer developed positively charged Poly (amidoamine) (PAMAM) dendrimer. An aptamer (Apt), which is sensitive to STX was electrostatically immobilized onto the PAMAM dendrimer layer. This results in an Apt that is preferably flat inside a Debye length, resulting in less charge-screening effect and a higher sensor signal. Capacitance-voltage and constant-capacitance measurements were utilized to monitor each step of a sensor surface variation, namely, the immobilization of PAMAM dendrimers, Apt, and STX. Additionally, the surface morphology of PAMAM dendrimer layers was studied by using atomic force microscopy and scanning electron microscopy. Fluorescence microscopy was utilized to confirm that Apt was successfully immobilized on a PAMAM dendrimer-modified EIS sensor. The results presented an aptasensor with a detection range of 0.5–100 nM for STX detection and a limit of detection was 0.09 nM. Additionally, the aptasensor demonstrated high selectivity and 9-day stability. The extraction of mussel tissue indicated that an aptasensor may be applied to the detection of STX in real samples. An aptasensor enables marine toxin detection in a rapid and label-free manner.

## 1. Introduction

Marine biological toxins accumulate in seafood through the aquatic food chain [1]. Depending on the toxin, they produce several types of poisoning, such as amnesic shellfish poisoning (ASP), diarrhetic shellfish poisoning (DSP), and paralytic shellfish poisoning (PSP) [2]. Toxicity can occur as a result of consumption of contaminated seafood, inhalation, or contact with skin; additionally, most marine toxins cannot be eliminated in high-temperature, acidic environments. PSP toxins, in particular, are strong neurotoxins produced by marine dinoflagellates [3,4]. Saxitoxin (STX) is a neurotoxin having a low molecular weight (299 g/mol) that belongs to the group of PSP toxins [5,6]. There is a significant risk to human health since STX may enter the food chain by accumulating in filter-feeding bivalves and fish [7,8]. Numbness, headaches, tingling, muscular weakness, respiratory failure, and even death may result from exposure to this toxin [8]. In most countries, the guideline concentration limit for STX in drinking water is 1–3 µg/L [5].

Mouse bioassay (MBA) [9], enzyme-linked immunosorbent assay (ELISA) [10], high-performance liquid chromatography (HPLC) [11], and liquid chromatography-mass spectrometry (LC-MS) [12] are the four primary methodologies and analyses [13] for the detection of marine toxins. For a long time, the MBA has been the mandated official technique for STX analysis [6,14]. It provides appropriate measures, but it has drawbacks such as limited specificity and sensitivity, and ethical concerns about the use of live animals [6,8,15]. Analytical approaches, such as HPLC and its related techniques, provide selective identification, separation, and sensitive quantification; nevertheless, they need highly competent and trained staff, toxic reference standards, costly and complex equipment, and time-consuming processes [7,8]. On the other hand, on-site detection is challenging owing to considerable pre-treatment, professional operation, and time-consuming concerns. Therefore, given the time-consuming nature of current approaches and the short reaction time of STX, robust analytical methods for the rapid and sensitive detection of STX are instantly needed. Electrochemical analysis with polyelectrolyte molecules might be a feasible solution to these challenges [16]. Nucleic acid and antigen-antibody coupled bioactive polyelectrolyte molecules have been designed for the detection of biomolecules with a suitable operation, excellent specificity, and high sensitivity [17].

Aptamers (Apts) are nucleic acid oligomers (DNA or RNA molecules) that have a high affinity and selectivity for a broad variety of particular target analytes [18]. Apts may be chemically generated in vitro, while antibodies need in vivo conditions [19]. Apts are a viable alternative for biosensing because of these and other characteristics, including high stability and the ease with which they may be changed by chemical processes to incorporate functional groups [18,20]. In the past 10 years, Apts targeting toxins have been identified, including ochratoxin A [21], okadaic acid [22], microcystin-LR [23], and even STX [14,15].

Poly (amidoamine) (PAMAM) dendrimer, like other dendrimer families, are three-dimensional nano-sized synthetic molecules with internal cavities and multiple surface groups that have the advantages of being non-toxic, biocompatible, having adequate functional groups for chemical fixation, having a small body accumulation, and being able to be utilized as a synthetic vector for a gene’s delivery [24,25]. Sensing applications have recently gained a lot of attention. Amine- or carboxyl-terminated dendrimers on their peripheral have the benefit of being able to conjugate to other molecules through an amide linkage, which is one of nature’s most basic and pervasive chemical linkages. For example, covalent amide linkages were used to connect amine-terminated PAMAM dendrimers to an activated mercaptoundecanoic acid self-assembled monolayer [26,27,28].

In this work, a layer-by-layer (LbL)-produced positively charged PAMAM dendrimer layer was used to modify an electrolyte-insulator-semiconductor (EIS) sensor for sensitive label-free detection of the STX. An EIS sensor is a biochemically sensitive capacitor that is simple to make and inexpensive. An adsorptively immobilized Apt will be preferably flat-oriented on a surface of EIS, with a negatively charged amide group directed to a positively charged PAMAM dendrimer molecules; due to a positively charged polyelectrolyte layer, both a Debye screening effect and an electrostatic repulsion between STX and Apt will be less effective, resulting in a higher detection signal. Capacitance-voltage (C-V) and constant-capacitance (ConCap) analyses were conducted to investigate variations in sensor capacitance caused by specific interactions of positively charged PAMAM dendrimer-modified Apt and STX. In theory, Apt conformational variations cause charge changes on a sensor surface, resulting in sensor capacitance variations. By monitoring variations in capacitances, a sensor was capable of detecting STX.

## 2. Materials and Methods

### 2.1. Chemical and Reagents

The 78-mer Apt 5′-end (5′ -GGT ATT GAG GGT CGC ATC CCG TGG AAA CAT GTT CAT TGG GCG CAC TCC GCT TTC TGT AGA TGG CTC TAA CTC TCC TCT-3′) [29,30] supplied by Sangon Biotech Shanghai Co., Ltd. (Shanghai, China), STX, dinophysistoxin (DTX), yessotoxins (YTXs), and pectenotoxins (PTXs) were bought from the National Research Council of Canada (NRC), and okadaic acid (OA) was provided by Cell Signaling Technology Co. Ltd. (Danvers, MA, USA). PAMAM dendrimers G-3 (molecular weight: 6848.79 g/mol) were bought from Weihai CY Dendrimer Technology Co., Ltd. (Weihai, China). Hydrofluoric acid (HF) was purchased from Sinopharm Chemical Reagent Co. Ltd. (Shanghai, China). The rest of the reagents were bought from Aladdin, Shanghai, China. Before utilization, Apt stock solution (100 µM) was synthesized using Milli-Q water and kept at −20 °C.

### 2.2. Sensor Fabrication

A sensor was made with minimal changes based on our earlier report [29,30]. As illustrated in Figure 1, an EIS structure of Au/n-Si/SiO_2_ was constructed on a silicon wafer (n-type, <100>, 10–15 Ωcm). To remove a SiO_2_ layer (30 nm) that was dry oxidized onto a silicon wafer as an insulating layer, HF was served to etch a rear side of a wafer. After that, a gold (Au) layer was placed on a rear side of a wafer. A wafer was ready for further experimentation after cutting it into tiny pieces of the desired size and cleaning it in an ultrasonic bath with acetone, isopropyl alcohol, ethanol, and deionized water (DI).

### 2.3. LbL Immobilization of PAMAM Dendrimers/Apt and STX

The LbL approach for electrostatic assembly of polyelectrolytes having alternating charges is easy, fast, effective, and minimally expensive [31,32,33]. The LbL technique was used to immobilize negatively charged Apt onto positively charged PAMAM dendrimer and to adsorb PAMAM dendrimer on the SiO_2_ surface gate insulator. Apt immobilized in LbL often forms flat, elongated structures [31]. As a result, low ionic strength solutions were used in this study, allowing STX to be positioned near a gate surface inside a Debye length, resulting in a higher sensor signal. This is crucial to improve the performance of the sensor since the charges of target molecules outside the Debye distance to the sensor surface could not generate responsive signals due to the screening effect. Before the adsorption of PAMAM dendrimers, first, a sensor surface was activated using a mixture of piranha solution (60 µL H_2_SO_4_ (98%) and 30 µL H_2_O_2_ (35%)) pipetted directly onto a sensor surface for at least 10 min at room temperature, then cleaned with DI water. Acid treatment was carried out three times. A sensor surface was then modified for 10 min with a 100 µL PAMAM dendrimer solution to generate the polyelectrolyte layer [29,34,35]. Sensor surfaces are likely to be sufficiently negatively charged to allow electrostatic adsorption of near-completely positively charged PAMAM dendrimer molecules, at pH 5.45 [36]. After a PAMAM dendrimer adsorption, a sensor was cleaned three times with a measurement solution to eliminate unbounded molecules from a SiO_2_ surface [36]. A 3 µL measure of 100 µM Apt was used to immobilize a PAMAM dendrimer-modified SiO_2_ sensor surface for around 1 h (h) at room temperature (RT), followed by cleaning a SiO_2_ surface with a measurement solution. For each test, a sensor surface coated with PAMAM dendrimer/Apt bilayer was incubated for 1 h at RT with various concentrations ranging from 0.1 nM to 100 nM of STX solutions. To remove unbound STX, a SiO_2_ surface was cleaned three times with a measurement solution after 1 h of incubation.

### 2.4. Electrochemical Measurements

For electrochemical measurements, an electrochemical workstation was used (Zennium, Zahner Elektrik, Bad Staffelstein, Germany). As illustrated in Figure 1, there is a three-electrode electrochemical measuring system that consists of Pt wire as a counter electrode, Ag/AgCl as a reference electrode, and an EIS sensor as a working electrode. A measuring solution was a low-ionic-strength solution (10 mM NaCl, pH 5.45) [36,37]. MPC227 pH/Conductivity Meter (Mettler-Toledo, Germany) was utilized to control the pH of all the solutions. The EIS sensor was calibrated for the detection of STX using a handmade measurement chamber. From bottom to top, a chamber was made up of an EIS surface (working electrode), a plastic container (box having a square hole at the bottom), and silicon rubber (seal ring). Variations in capacitance were investigated using C–V and ConCap analysis. For C-V analysis, a direct current (DC) gate voltage (−0.5 V to + 1.5 V, 100 mV increments) and a low superimposed alternate current (AC) voltage (20 mV, 60 Hz) were used. In order to obtain a ConCap analysis, gate voltage was served to use a feedback control circuit under a fixed capacitance calculated by C-V analysis [36,38]. Entire measurements were taken at RT. A Faraday box was served to protect a measuring chamber from the influence of an electromagnetic field. During the stability test, the sensors were rinsed with saline solution, had excess water removed by filter paper, and were placed in a 4 °C refrigerator between each measurement. Three sensors were used during the whole test for repeatability. Mussels were acquired fresh from a local market (Xi’an, China) for real sample preparation. Treated and untreated mussels (digestive glands removed) were separated from the shells and cleaned with DI water, as reported in the literature [29]. Then, 2 mL (50%) methanol was introduced to (0.5 g) mussel tissue, which was vortexed vigorously for 5 min. The supernatant was collected using centrifugation. A solution was then heated to 75 °C for 5 min before being centrifuged at 4000 rpm for another 5 min. The supernatant (pH 5.45, adjusted by HCl) was collected and kept at 4 °C, which could be applied to analyze a real sample. A certain amount of STX was added to untreated mussels and extracts as detection solutions. The electrochemical test was carried out similarly to the previous process.

### 2.5. Characterizations Methods

Sensor surface morphology following PAMAM dendrimer and Apt immobilization was studied utilizing a Hitachi SU-70 scanning electron microscope (SEM). A scanning probe microscope (SPM^_^9700HT, SHIMADZU, Kyoto, Japan) with Au-coated cantilevers was utilized to characterize the samples using atomic force microscopy (AFM). All images were scanned at a resolution of 2 µm × 2 µm, with a scan speed of 1 Hz. The frequency of resonance varies between 230 to 380 kHz (Nanosensors, Switzerland). Following PAMAM dendrimer and Apt immobilization, the morphology of a surface and the roughness of a sensor surface were investigated. Fluorescence analyses were performed using an Axio Imager A1 m (Zeiss Axio Imager, Suzhou, China) equipped with an appropriate filter setup.

## 3. Results and Discussion

### 3.1. Characterization of Biosensor Preparation

After PAMAM dendrimer and Apt immobilization on an EIS sensor surface, SEM analysis was used to analyze sensor surface morphology, as illustrated in Figure 1. Figure 1a displays an SEM image of a bare SiO_2_ sensor surface, whereas Figure 1b demonstrates a white spherical dot shape after PAMAM dendrimers are immobilized on a bare sensor surface. As illustrated in Figure 1c, the assembled nanoflakes were seen after Apt was immobilized on a PAMAM dendrimer-modified sensor surface. The surface roughness was determined by utilizing a root mean square value (RMS) and surface area different from AFM height images. Figure 2 presents AFM images of the bare sensor surface, PAMAM dendrimer layer, and PAMAM dendrimers/Apt. A bare sensor surface seems to be smooth, with an average RMS value of 1.04 nm (Figure 2a). After PAMAM dendrimer and Apt immobilization, AFM images display apparent changes when compared to a bare sensor surface. Figure 2b illustrates that surface roughness increases when a PAMAM dendrimer molecule is adsorbed (RMS = 3.44 nm) on a bare sensor. AFM images of a PAMAM dendrimer layer exhibited a uniform distribution across a sensor surface, implying that the PAMAM dendrimer has a monodispersed spherical confirmation, having a highly branched three^-^dimensional structure that serves as a scaffold for multiple biomolecule attachment [39]. In addition, in an AFM image of a PAMAM dendrimer layer, surface morphology changed significantly following Apt immobilization on a PAMAM dendrimer-modified sensor surface (Figure 2c). Huge clusters seem to occupy a surface of a PAMAM dendrimer/Apt bilayer, raising surface roughness to RMS = 19.63. Hence, AFM characterization analysis validates an effective formation of a PAMAM dendrimer/Apt bilayer on the surface of a sensor, which is in accordance with our previous studies [30].

### 3.2. Confirmation of Apt Immobilization

Fluorescent analysis was served as a reference method to validate an immobilization of Apt utilizing an Axio Imager A1m (Zeiss Axio Imager, Suzhou, China) fluorescence microscope with applicable filter configuration. To observe a successful Apt immobilization onto a PAMAM dendrimer layer, fluorescence dye^_^6^_^carboxyfluorescein-labeled Apt (FAM-labeled Apt) was utilized. FAM-labeled Apt was first introduced to a PAMAM dendrimer-modified EIS sensor to confirm the hybridization method. Figure 3 demonstrates fluorescence measurements after applying the FAM-labeled Apt solution (5 µM) to a bare sensor surface and a PAMAM dendrimer-modified EIS sensor surface, also after applying an STX solution (5 µM) to an EIS sensor modified with a PAMAM dendrimer/Apt bilayer. Figure 3a illustrates that no fluorescence signal was detected when an FAM-labeled Apt solution was applied to an EIS sensor. Electrostatic repulsion between an Apt and a SiO_2_ sensor surface hinders the immobilization process because both have negatively charged. Therefore, no FAM-labeled Apt stays on a sensor surface following a rinsing procedure. A strong and uniform fluorescence signal was acquired after incubation of an FAM-labeled Apt solution onto a PAMAM dendrimer-modified EIS sensor surface as displayed in Figure 3b, presenting successful Apt immobilization onto a PAMAM dendrimer layer. Even after six cleaning processes, a fluorescence signal was observed, without any reduction in fluorescence intensity. In contrast to Figure 3b, Figure 3c illustrates a weaker fluorescence signal after STX was attached to FAM-labeled Apt. These analyses validate that an FAM-labeled Apt was effectively adsorbed onto a PAMAM dendrimer layer.

### 3.3. Real-Time Detection of Saxitoxin

After STX was attached to an EIS sensor surface, changes in sensor capacitance were detected via shifts in C-V curves and potential variations in ConCap mode, resulting in variations in surface charges generated by an attachment of charged molecules, as in Figure 4. The C-V curve of a bare sensor was modified by immobilization of PAMAM dendrimers, Apt, and STX, as illustrated in Figure 4a. Typical high-refinery forms of accumulation, depletion, and inversion regions were observed. In this work, C-V curves (Figure 4a) and the ConCap response (Figure 4b) were obtained in a measuring solution at pH 5.45 before and after adsorption of PAMAM dendrimers, Apt, and STX. As can be seen, an optimal capacitance in an accumulation area of a C-V curve stays almost constant during surface variation stages, which is in accordance with previous studies utilizing a capacitive EIS sensor to detect marine biological toxins [29,30]. Significant shifts in C-V curves across a voltage axis have been seen in the depletion zone, having the direction and magnitude of the shifts changing depending on the sign and amount of an adsorb charge. This demonstrates that charged macromolecule adsorption and interaction result in an interfacial potential shift, resulting in a variation in flat band voltage and capacitance of the EIS structure [8]. The depletion layer is expended by an attachment of positively charged PAMAM dendrimers to a negatively charged bare SiO_2_ surface, lowering a space–charge capacitance in S_i_ and a variable space-charge capacitance of a semiconductor (C_sc_). Resultantly, a sensor’s total capacitance will decrease, and a C-V curve will shift toward a larger negative gate voltage. After Apt binding, a shift in the C-V curve toward high voltage was seen. The C-V curve shifted much farther to a higher voltage direction of a gate voltage when an STX was introduced to a PAMAM dendrimer-modified sensor. STX and Apt interact to cause Apt conformational changes and charge redistribution on a SiO_2_ sensor surface. Resultantly, gate voltages on a C-V curve have shifted to a higher voltage. In order to provide real-time monitoring of potential shifts produced by changes in surface charge, a ConCap measuring method was applied. Figure 4b demonstrates dynamic ConCap analysis data obtained after adsorption of PAMAM dendrimers, Apt, and STX. It was observed that a ConCap signal was reduced when adsorption of PAMAM dendrimers was added. Increasing in a positive potential shift was observed after immobilization of an Apt, which was caused by charge shifts produced by an Apt change on the sensor surface. Moreover, when STX was immobilized, a conformation modification and charge rearrangement led to further potential changes in the direction of the higher voltage.

### 3.4. STX Detection

In the STX detection tests, several concentrations of STX ranging from 0.1 nM to 100 nM were utilized. Figure 5a depicts a zooming-in of C-V curves in a depletion zone of different STX concentrations measured in a depletion region. As discussed earlier, curves moved in a positive direction, resulting in an STX being attached to them. Moreover, when a higher concentration of STX was used, a higher change was seen. ConCap findings revealed higher potential changes along with higher STX concentrations, which were consistent with C-V studies (Figure 5b). In accordance with statistical data, STX demonstrated a linear response to potential variations in a concentration range of 0.5 nM to 100 nM. The following equation was developed to represent an association between potential shifts and STX concentrations: Figure 5c shows the potential shift (V) = 0.1755 + 0.0983 × log [C_STX_], with R^2^ = 0.926. The detection limit of the PAMAM dendrimer-modified Apt (S/N = 3) for the STX was as low as 0.09 nM.

### 3.5. Sensor Selectivity and Stability

This aptasensor was utilized to test the selectivity of the other four DSP toxins, which were chosen from a category that included DTX-1, PTX, OA, and YTX. Aptasensor reacted to STX (10 nM), and the responses to the other toxins (100 nM) were recorded, with NaCl (100 nM) serving as a negative control Figure 6. The response to STX was much higher when compared to those to DTX-1, PTX, OA, and YTX, demonstrating that it has strong selectivity. It may be possible that this is because STX-Apt does not recognize DTX-1, PTX, YTX, or OA, and thus the sensor based on an STX–Apt is not sensitive to DTX-1, PTX, OA, and YTX. The results suggest that biosensor response is selective to STX, demonstrating that a biosensor has a high selectivity for the detection of STX. To evaluate the stability of the aptasensor, the aptasensor was kept at 4 °C for 15 days. The aptasensor reaction remained constant for the first 9 days, then steadily declined (see Figure 6a). Prolonged storage may degrade or oxidize, causing the potential shift to be reduced.

Table 1 highlights and summarizes most of the label^-^free STX aptasensors that have been reported [6,8,40,41,42]. Our designed aptasensor has a much lower LOD with a significantly extensive detection range, which is significant. Moreover, since this procedure is easy to use, it has the potential to be utilized in a variety of applications, including label-free detection of toxins and assessment of environmental pollution.

In a real^-^sample test, this aptasensor was evaluated using a mussel tissue extraction method. STX was introduced to mussel tissue extraction, and EIS measurement was performed again. The rate of recovery (%) was determined as follows: ∆P_mussel tissue extraction_/∆P_buffer_ (∆P = potential shift) was utilized to calculate a rate of recovery (%). In order to determine if treatment of mussels (digestive glands removed) results in an elimination of toxins, we also conducted experiments on untreated samples. In order to accomplish this, the toxin was introduced to a sample before treatment. The results of the measurements are presented in Table 2. In mussel tissue extraction, recovery rate for the aptasensor was 101.4% (C_STX_ = 100 nM). Additionally, it is stated that there are no statistically significant differences in the recovery rate of treated and untreated samples. This demonstrates that the treatment of real samples did not cause significant losses of the toxins used in the experiment. It is also shown that this aptasensor has high stability and recovery when used for the detection of real samples. According to all of the results, a proposed aptasensor could be utilized to evaluate real-sample tests. In principle, aptamers may exactly attach to STX, causing conformational variations and charge rearrangement. These two aspects were capable of producing a change in surface charge in PAMAM and an insulating layer in PAMAM/Apt, which was described as a variation in capacitance (Figure 1). Figure 4 illustrates that, after binding to STX, which was also used in a ConCap analysis, the gate voltages of the C-V curves shifted in a positive direction. As STX concentrations increased, the effect increased. As a result, it was found that the concentrations and the ConCap results had a linear relationship, suggesting that a detection range was linear. The systematic evolution of the ligand exponential enrichment (SELEX) method was utilized to demonstrate that aptamers have a high affinity for their targets [41]. Despite the fact that the structures of DTX-1, PTX, OA, and YTX are quite similar to those of STX, aptamers and toxins interact very differently. Due to a considerable interaction between STX and Apt, the confirmation of the latter changed, resulting in charge rearrangement and clear potential shift. Nevertheless, because of weak interactions between DTX-1, PTX, OA, and YTX, the possibility of a shift was negligible.

## 4. Conclusions

In summary, STX was detected using an EIS sensor composed of Au-n-Si-SiO_2_. An LbL technique was utilized to electrostatically adsorb PAMAM dendrimers onto the SiO_2_ layer also to easily and rapidly Apt immobilize onto a PAMAM dendrimer layer. A benefit of the adsorptive immobilization approach is that both Apt and STX are attached near the EIS surface with molecular charges situated inside the Debye length of a gate surface, leading to a larger sensor signal. Aptasensor exhibited excellent linear responses to STX at concentrations ranging from 0.5 nM to 100 nM. Aptasensor has a LOD of 0.09 nM, which was equivalent to earlier reported methods. Additionally, the aptasensor displayed a high degree of specificity and stability, which is beneficial for its potential applications. STX detection with an excellent recovery rate in mussel tissue extraction displayed that an aptasensor may be employed for real-time sample detection. Therefore, the aptasensor offers a low-cost, label-free, and efficient method for detection of STX, with high specificity, a low LOD, and high selectivity, enabling novel applications in a variety of fields associated with marine toxin detection, such as food quality control and water.

## Data Availability

The data presented in this work are available on request from the corresponding authors.

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
