# Peer review of "Label-Free Detection of Saxitoxin with Field-Effect Device-Based Biosensor"

_nanomaterials, 2022, doi:10.3390/nano12091505_

Round 1

Reviewer 1 Report

I review the manuscript “Label-Free Detection of Saxitoxin with Field-Effect Device-Based Biosensor” (Nanomaterials). The article idea is interesting and the use of nanometric sensors (including synthesis, characterization and application) fits with the journal scope.

In my opinion it is required some additional experiments to support the ideas stated.

Some considerations:

Line 23-24. The linear interval stated is 0.5 - 100 nM, however the limit of detection is 0.03 nM. I assume the limit of quantification of 0.09 nM (please verify your validation protocol). It seems to be a mistake in the analytical parameters calculation.

Line 124. Why is important the Debye distance?, how does it affects the sensor performance? Please include an explanation about the effect of the experimental conditions in the distance reduction.

Line 186. Please evaluate the precision of the methodology. It cannot be enough for centesimal values (i.e. 1.04 nm)

Line 187-198. Is there described the morphological effect in other systems?, I suggest to include a reference.

Line 240. Which is the pH value of the system? Is it necessary to evaluate the effect of the pH value?, please include an explanation.  

Line 288. What is the specific interaction between the aptamer and STX?

Line 309. The authors evaluate the accuracy through recovery test; however it is mentioned that there is an official methodology. If there is an official methodology, accuracy must be evaluated by comparison of the results obtained with the proposed and the official methodologies.

Reviewer 2 Report

The manuscript by N. Ullah et al., titled “Label-Free Detection of Saxitoxin with Field-Effect Device-Based Biosensor” reports on the aptamer-based sensor for the detection of one of the paralytic shellfish toxins – saxitoxin. Development of novel sensors and assays for toxin detection is an interesting topic. However, novelty of this work is somewhat limited as it is very similar to the previous work by the same authors https://doi.org/10.3390/s21144938.

Comments.

  • Introduction. This section needs to be clarified and re-organized. Saxitoxin can be produced by both cyanobacteria and microalgae, the former can result in the water contamination and the latter in the toxin accumulation in the bivalves. As the authors apply developed sensor to the STX detection in mussel extract, it is the second situation that needs to be reflected in the literature review, i.e. saltwater HABs and toxin accumulation in bivalves.
  • Ref. 4 describes intoxication due to water contamination with other cyanobacterial toxins - microcystins, not saxitoxin. There are several recorded cases of human intoxication by saxitoxin and its analogues resulting from the consumption of contaminated seafood, and they should be cited instead as more relevant to this work.
  • Line 145. Measurements with sensor were in the 10 mM NaCl solutions with pH 5.45. How was pH adjusted?
  • Fig. 6 (a). What was the reason to test selectivity of the sensor to other marine toxins that have structures very different from STX and, thus, could not be expected to be detected by the aptamer? Was selectivity to other paralytic shellfish toxins that always occur concomitantly with STX tested?
  • Fig. 6 (b). How bound STX was eliminated from the sensor after measurements? How many times these procedures could be repeated be used without losing its sensitivity?
  • Mussel extracts were prepared using 50% methanol aqueous solution and calibration measurements with the sensor were carried out in NaCl aqueous solutions with pH 5.45. Did presence of methanol interfere with the measurements with the senso? If it did not, why calibration measurements were not made in the water/methanol solutions as well? The authors stated that pH of the solution is important for the formation of the sensitive layer. What was pH of the mussel extracts and how was it controlled?
  • Procedure of STX detection in mussel extract should be described in more details. How STX concentration in the mussel tissue was determined? Was sensor calibration in standard solutions used for this purpose? It appears from the presented results that sensor was calibrated in the mussel extract spiked with STX. In this case, it is not correct to talk about recovery rates, what can be discussed is the sample matrix effect on the sensor response. Please, comment.

Reviewer 3 Report

This manuscript describes the characterization of a Field-Effect biosensor based on layer-by-layer nanostructures for saxitoxin detection. The novelty, significance of the work, and the data presented in this manuscript are all satisfactory. Physicochemical analysis of the device seems to be complete. Since the important character of the present work, the paper can be recommended for publication in Nanomaterials.

Minor comment:

Please verify the data regarding linear range (0.5 – 100 nM), limit of detection (0.3 nM) and limit of quantitation. It seems that the value of LOQ will be in the range of linearity which should be corrected. LOD and LOQ should both be below linear range.

Round 2

Reviewer 1 Report

The authors considered the suggestions proposed. In my opinion, the manuscript can be accepted.

Author Response

Thank you for the very posive response.

Reviewer 2 Report

The authors have addressed most of the comments and the manuscript has improved. However, an important question about functioning of the developed sensor has not been answered.

It concerns removal of STX from the sensor, reuse of the sensor and repeatability of the sensor response. The authors wrote in the answer that STX bounds strongly to aptamer and cannot be removed without damaging the aptamer. Does it mean that every experimental point was measured using new sensor? For instance, Fig. 6 b shows stability of the sensor in 10 nM STX solutions during 15 days with the average of three measurements shown for each day. From the authors’ explanation it appears that STX could not be removed from the sensor, thus, sensor responded once, and this figure simply shows stability of the aptamer layer with bound STX?

The authors should explain in detail how calibration measurements in standard STX solutions and mussel extracts were carried out.
